# Human Cytomegalovirus miR-UL70-3p Downregulates the H_2_O_2_-Induced Apoptosis by Targeting the Modulator of Apoptosis-1 (MOAP1)

**DOI:** 10.3390/ijms23010018

**Published:** 2021-12-21

**Authors:** Abhishek Pandeya, Raj Kumar Khalko, Anup Mishra, Nishant Singh, Sukhveer Singh, Sudipta Saha, Sanjay Yadav, Sangeeta Saxena, Sunil Babu Gosipatala

**Affiliations:** 1Department of Biotechnology, Babasaheb Bhimrao Ambedkar University, Lucknow 226025, India; abhi12pandey@rediffmail.com (A.P.); rajkumar2010khalko@gmail.com (R.K.K.); anupmishra644@gmail.com (A.M.); dr_sangeeta_saxena@yahoo.com (S.S.); 2Developmental Toxicology Division, CSIR-Indian Institute of Toxicology Research (CSIR-IITR), Lucknow 226001, India; nishantbst@gmail.com (N.S.); sukhveersingh5@iitr.res.in (S.S.); sanjayitrc@gmail.com (S.Y.); 3Department of Pharmaceutical Sciences, Babasaheb Bhimrao Ambedkar University, Lucknow 226025, India; sudiptapharm@gmail.com; 4Department of Biochemistry, All India Institute of Medical Sciences, Rae Bareli 229405, India

**Keywords:** apoptosis, hcmv-miR-UL70-3p, modulator of apoptosis-1 (MOAP1)

## Abstract

Human Cytomegalovirus (HCMV) is a prototypic *beta herpesvirus*, causing persistent infections in humans. There are medications that are used to treat the symptoms; however, there is no cure yet. Thus, understanding the molecular mechanisms of HCMV replication and its persistence may reveal new prevention strategies. HCMV evasive strategies on the antiviral responses of the human host largely rely on its significant portion of genome. Numerous studies have highlighted the importance of miRNA-mediated regulation of apoptosis, which is an innate immune mechanism that eradicates virus-infected cells. In this study, we explore the antiapoptotic role of hcmv-miR-UL70-3p in HEK293T cells. We establish that hcmv-miR-UL70-3p targets the proapoptotic gene Modulator of Apoptosis-1 (MOAP1) through interaction with its 3’UTR region of mRNA. The ectopic expression of hcmv-miR-UL70-3p mimic significantly downregulates the H_2_O_2_-induced apoptosis through the translational repression of MOAP1. Silencing of MOAP1 through siRNA also inhibits the H_2_O_2_-induced apoptosis, which further supports the hcmv-miR-UL70-3p mediated antiapoptotic effect by regulating MOAP1 expression. These results uncover a role for hcmv-miR-UL70-3p and its target MOAP1 in regulating apoptosis.

## 1. Introduction

Human Cytomegalovirus (HCMV) is a ubiquitous human pathogen, exhibiting an estimated seroprevalence of about 60 to 90% around the world [1]. Following upon primary infection, it causes latent infections in the immunocompetent hosts without much clinical manifestations, whereas it causes significant morbidity and mortality in immune-naive and immunocompromised individuals [2]. It contains linear DNA of ≈230 Kb as a genome, encoding >170 proteins and numerous long and small non-coding RNAs [3]. Being a herpesvirus family member, it causes latent infections in the host and evolved many strategies to escape the antiviral arsenal of human host, and a significant portion of the genome is dedicated for this purpose. The proteins, RNA, and miRNAs of HCMV are reported to evade the antiviral responses such as apoptosis by targeting and modulating the various proteins involved in apoptotic pathways [4,5,6,7,8,9,10,11,12,13]. Recent literature suggests the pivotal role(s) of HCMV miRNAs in regulating the apoptosis in such a way that the outcome should be beneficial to the virus. HCMV encodes 26 mature miRNAs (miRbase, ver. 22.1; October 2018), spanning throughout the genome, which is expressed during the lytic as well as latent phases of infection [3] and identified to target both host [14,15] and viral mRNA transcripts [14,16]. 

Apoptosis is an innate antiviral response that eliminates the viral-infected cells from the body, thereby preventing the viral infection and propagation. Many viruses reported to modulate this process including the HCMV. Inhibition of apoptosis in the early phase of infection is beneficial to HCMV, as it happens to be a slow replicating virus (requires 48–72 h for producing progeny), so all its products regulate the onset of apoptosis [17]. Multiple studies reported the regulation of apoptosis by HCMV miRNAs, and the first report came from the studies of Lee et al. in the year 2012, where the hcmv-miR-UL112 reported to inhibit the apoptosis by translational repression of Bcl-2 Associated Transcription Factor1 (BclAF1) [8]. Later, Wang et al. in 2013 showed that the ectopic expression of hcmv-miR-UL148D inhibits the Immediate Early Gene X1 (IEX1) induced apoptosis by targeting and downregulating the mRNA of IEX1 in HEK293 cells [9]. By performing *in silico* studies, we also speculated the antiapoptotic activity of hcmv-miR-UL70-3p [10]. In continuation, another HCMV miRNA, hcmv-miR-UL36-5p, was reported to inhibit apoptosis by the translational repression of Adenine Nucleotide Translocator 3 (ANT3/ SLC25A6) in HEK293 cells [11]; the studies following the AGO-CLIP-Seq-based miRNA targetomics approach reported antiapoptotic activity to the HCMV miRNAs. The HCMV miRNAs, miR-US4-3p, miR-US5-1, miR-US5-2-3p, miR-US25-2-3p, miR-UL22A-3p, miR-UL36-3p, and miR-UL112-5p were identified to target and downregulate the apoptotic genes FAS, FADD, Caspase 3 and Caspase 7 [12]. Recently, Hancock et al. (2021) reported the antiapoptotic activity of hcmv-miR-US5-1 and miR-UL112-3p. They exhibit this activity by targeting and downregulating the Forehead Box O3a (FOXO3a) transcript, and its protein in hematopoietic progenitor cells results in reducing the expression of proapoptotic gene BCL2L11, preventing virus-induced apoptosis [13]. 

The existence of hcmv-miR-UL70-3p was first mentioned in the studies of Grey et al. in 2005 as miR-UL70-1 [18], and the miR-UL70-5p was reported to found in the dense bodies produced during viral infections [19]. The studies of Naqvi et al. show that this miRNA alters the cellular miRNA expression profiles in human oral keratinocytes and predicted to interfere the apoptosis, chemokine, endocytosis, Erb pathway, and osteoclast differentiation pathways [20]. The expression and kinetics of this miRNA during lytic replication in fibroblasts suggest it may play a role in maintaining the quiescent state [21]. The studies of Ulasov et al. found that this miRNA promotes glioblastoma multiforme (GBM) cancer stem cells stemness [22]. In the present study, we evaluated our earlier *in silico* findings related to the antiapoptotic nature of hcmv-miR-UL70-3p in HEK293T cells by inducing the apoptosis through hydrogen peroxide (H_2_O_2_). 

## 2. Results

### 2.1. Effect of hcmv-miR-UL70-3p on H_2_O_2_ Induced Apoptosis

To examine the effect of hcmv-miR-UL70-3p on apoptosis, we induced apoptosis in HEK293T cells through H_2_O_2_ as described by Xiang et al. [23] and described in the Materials and Methods. The morphological changes characteristic to apoptosis, including cell shrinkage, membrane blebbing, chromatin condensation, phosphatidylserine flippage, and measurement of Caspases 3/7 was studied through DAPI staining, scanning electron microscopy, flow cytometry, and Caspase 3/7 assay. The HEK293T cells were divided into four groups: 1st group, negative control (no H_2_O_2_ treatment), 2nd group, positive control (H_2_O_2_ treated), 3rd group, transfected with hcmv-miR-UL70-3p followed by H_2_O_2_ treatment, and 4th group, transfected with hcmv-miR-UL70-3p and its sequence-specific inhibitor, followed by H_2_O_2_ treatment.

#### 2.1.1. DAPI (4’,6-Diamidino-2-phenylindole) Staining

The different experimental cell groups mentioned above were evaluated for chromatin condensation and nuclear fragmentation through DAPI staining. The images were captured at 63× magnification by using a confocal microscope (Figure 1(a1–a4)). The chromatin condensation and nuclear fragmentations were reduced in the group of cells transfected with hcmv-miR-UL70-3p (Figure 1(a3)) compared to the H_2_O_2_-treated cells (Figure 1(a2); ***, *p* < 0.001). Blocking the hcmv-miR-UL70-3p through its sequence-specific inhibitor shows the increment in the chromatin condensation and nuclear fragmentation compared to the cell group treated with hcmv-miR-UL70-3p alone (Figure 1(a4); ***, *p* < 0.001), suggesting that the hcmv-miR-UL70-3p downregulated the chromatin condensation and nuclear fragmentation, inferring the decrement of apoptosis. The apoptotic rate in these cell groups was assessed by using the ImageJ software, and it was found to be 2.48% in the negative control group without treatment of H_2_O_2_ and or hcmv-miR-UL70-3p, 20.68% in the cell group treated with H_2_O_2_, and 6.51% in the cell group transfected with hcmv-miR-UL70-3p mimic along with H_2_O_2_ (Figure 1d). A decrement of 14.17% of apoptotic rate was observed by the hcmv-miR-UL70-3p; further, this antiapoptotic rate was increased to 11.64% when we abrogate the hcmv-miR-UL70-3p with its inhibitor (Figure 1d). The apoptotic rate was calculated from the triplicate images of all the groups by employing one-way ANOVA (Appendix A) 

#### 2.1.2. Scanning Electron Microscopy Analysis for Membrane Blebbing

The above-mentioned experimental groups of cells were analyzed for the morphological variations characteristic to apoptosis such as membrane blebbing/ruffling through scanning electron microscopy (SEM). The images were captured at 800× (Figure 1(b1–b4)) and 4000× (Figure 1(b5–b8)) magnifications using the scanning electron microscope. The membrane blebbing is high in the group of cells treated with the H_2_O_2_ only compared to the group of cells without any treatment (Figure 1(b2, b6)), whereas it was found decreased in the group of cells treated with hcmv-miR-UL70-3p (Figure 1(b3, b7)). The abrogation of the hcmv-miR-UL70-3p activity through its inhibitor shows the increment in membrane blebbing (Figure 1(b4, b8)). The mean cell surface area increment and decrement for the SEM images (800×) were assessed through ImageJ software, which generated their 3D surface plots. Based on these surface plots, the histograms were generated, and the mean coverage area percentage for the H_2_O_2_ group of cells was found to be 33.95, whereas in the group of cells treated with hcmv-miR-UL70-3p, it was increased to 71.85, suggesting an increment in the mean coverage area, which resulted in the inhibition of apoptosis (Figure 1e). 

#### 2.1.3. Evaluation of Apoptosis through Flow Cytometry

In order to verify whether the hcmv-miR-UL70-3p could downregulate the H_2_O_2_-induced apoptosis or not, we performed flow cytometry analysis in the said experimental groups using annexin V and propidium iodide staining. The dot plots obtained were arranged in the lower panel; i.e., Figure 1c. The percentage of apoptotic cells in the group of cells treated with H_2_O_2_ was found to be 11%; however, the hcmv-miR-UL70-3p transfected group of cells shows the decrement to 3.7%. When we block the hcmv-miR-UL70-3p with its inhibitor, apoptotic cells are increased to 8.3%, which is statistically significant (**, *p* < 0.01; Figure 1f). 

#### 2.1.4. Measurement of Caspases 3/7 Activity

The effector caspases 3/7 activity in the said experimental groups of cells was measured through Caspase Glo 3/7 assay, and the results obtained are shown in Figure 1g. The caspase activity in the cells was measured as raw luminescence units (RLU), and it was found to be 5.87 × 10^5^ in the group of cells treated with H_2_O_2_, whereas it found to decrease to 2.35 × 10^5^ in the group of cells group transfected with hcmv-miR-UL70-3p; a decrement of 3.52 × 10^5^ in RLU is observed, which is statistically significant (****, *p* < 0.0001). Furthermore, we evaluated whether the transfection of hcmv-miR-UL70-3p alone can induce any Caspase 3/7; levels were checked, and we found that the RLU value is 0.410 × 10^5^, which is less when compared to the RLU obtained through H_2_O_2_ treatment. Blocking the action of hcmv-miR-UL70-3p with its inhibitor, then, the RLU was enhanced to 5.54 × 10^5^ (****, *p* < 0.0001), suggesting that the hcmv-miR-UL70-3p downregulates the expression of Caspase 3/7, inferring that it decreases the apoptosis. 

### 2.2. MOAP1 Is a Direct Target of hcmv-miR-UL70-3p

#### 2.2.1. Hcmv-miR-UL70-3p Regulates the Expression of MOAP1 mRNA

The expression of hcmv-miR-UL70-3p mimic in the different groups of cells transfected with and without miR-UL70-3p mimic, miR-UL70-3p mimic along with its inhibitor were analyzed through qRT-PCR using the specific primers as mentioned by Shen et al. 2014 [21]. The results (Figure 2a) show that the hcmv-miR-UL70-3p was expressed significantly in the group of cells transfected with miRNA mimic when compared to the control cells without transfection, while its expression was decreased in the group of cells transfected along with its specific inhibitor (***, *p* < 0.001). Then, the mRNA expression studies show that H_2_O_2_ treatment induced the MOAP1 mRNA when compared to the control group of the cells (Figure 2b). The ectopic expression of hcmv-miR-UL70-3p significantly downregulated the expression of MOAP1 mRNA (Figure 2b). Further, blocking the hcmv-miR-UL70-3p increased the MOAP1 mRNA expression, suggesting that the hcmv-miR-UL70-3p downregulates the MOAP1 mRNA expression. 

#### 2.2.2. Hcmv-miR-UL70-3p Binds the 3’UTR of MOAP1

The above results support our earlier *in silico* predictions that hcmv-miR-UL70-3p targets 3′UTR of MOAP1 mRNA (single binding site at position 527 to 549). So, we verified this binding by using dual luciferase vector constructs containing 3’UTR of MOAP1. The dual luciferase expression vector pEZX-MT06 containing the whole 3’UTR of MOAP1 (pEZX-MT06-3’UTR^WT^-MOAP1) and deleted 3’UTR of MOAP1 (pEZX-MT06-3’UTR^DEL^-MOAP1) and transfected them in the 5 different groups of cells, as mentioned in the Materials and Methods. The cartoon depicting the pEZX-MT06 dual luciferase vector is shown in Figure 2c. Each of these vector constructs were co-transfected into the cells along with the hcmv-miR-UL70-3p and/or its inhibitor, as shown in the Figure 2d. The relative luciferase activity (RLA) was measured through the luminometer and plotted as percentage RLA. The group of cells co-transfected with the hcmv-miR-UL70-3p and pEZX-MT06-3’UTR^WT^ -MOAP1 showed a significant decrease in relative luciferase activity in comparison to the group of cells co-transfected with hcmv-miR-UL70-3p and pEZX-MT06-3’UTR^DEL^ -MOAP1 (***, *p* < 0.001) (Figure 2d). Meanwhile, in the group of cells co-transfected with the hcmv-miR-UL70-3p and pEZX-MT06-3’UTR^DEL^-MOAP1, the luciferase activity was not affected. This result suggests the functionality of the binding site in the 3’UTR of MOAP1 for the hcmv-miR-UL70-3p. 

#### 2.2.3. Hcmv-miR-UL70-3p Downregulates the MOAP1 Protein Expression

To address the question of whether the binding of hcmv-miR-UL70-3p to the 3’UTR of MOAP1 mRNA results in the translational repression of MOAP1 protein or not was assessed through Western blotting. The MOAP1 protein expression was analyzed in the group of cells treated with or without H_2_O_2_, hcmv-miR-UL70-3p with H_2_O_2_, and hcmv-miR-UL70-3p along with its inhibitor and H_2_O_2._ The cell group transfected with hcmv-miR-UL70-3p mimic substantially downregulated the MOAP1 levels (Figure 2e). The abrogation of hcmv-miR-UL70-3p with its inhibitor, the MOAP1 protein, was upregulated. The relative protein quantification was performed with ImageJ software after normalizing the MOAP1 levels with β-actin. The relative MOAP1 levels were downregulated by the hcmv-miR-UL70-3p (*, *p* < 0.05), and the downregulation was inhibited when we block the hcmv-miR-UL70-3p through its inhibitor (**, *p* < 0.01) (Figure 2f). These results demonstrate that hcmv-miR-UL70-3p translationally represses the MOAP1 protein and thereby downregulates the apoptosis.

### 2.3. Hcmv-miR-UL70-3p and siRNA of MOAP1 Effect on the Expression of MOAP1 mRNA and Its Protein

The above studies confirm the downregulatory effect of hcmv-miR-UL70-3p on MOAP1 mRNA and its protein. We compared this effect of hcmv-miR-UL70-3p with the small interfering RNA (siRNA) designed against the 3’UTR of MOAP1 (38–66 position of the 3′UTR of MOAP1). The HEK293T cells were transfected with either hcmv-miR-UL70-3p (25 nM) or siRNA of MOAP1 (25 nM) and induced apoptosis with H_2_O_2_. The apoptotic inhibition, MOAP1 mRNA, and protein downregulations between them were compared through DAPI analysis, Caspase 3/7 assay, qRT-PCR, and Western blotting. The DAPI analysis (Figure 3(a1–a4)) and Caspases 3/7 activity (Figure 3b) show that both downregulated the chromatin condensation/nuclear fragmentation and Caspase 3/7 activity in comparison to the control; however, the siRNA of MOAP1 exhibits higher downregulation than hcmv-miR-UL70-3p. The MOAP1 mRNA (Figure 3c) and its protein (Figure 3d) downregulations were plotted, and the siRNA of MOAP1 shows more downregulation on MOAP1 mRNA in comparison with the hcmv-miR-UL70-3p (±SEM; **, *p* < 0.01); however, there is not much difference in the protein downregulation between them. 

## 3. Discussion

The present study demonstrates the antiapoptotic activity of hcmv-miR-UL70-3p by translational repression of the proapoptotic gene MOAP1. The ectopic expression of hcmv-miR-UL70-3p downregulates the H_2_O_2_-induced apoptosis in HEK293T cells, which is assayed through microscopy, Caspase 3/7 assay, and flow cytometry. The results of these assays confirm the downregulatory effect of hcmv-miR-UL70-3p on apoptosis; however, the extent of downregulation varies from one assay to the other. The apoptotic downregulation by the hcmv-miR-UL70-3p is found to be 14.17% through the DAPI staining method, 37.89% through SEM analysis, and 7.47% through flow cytometry, and it is about 59.96% through Caspase 3/7 assay. The percentage of inhibition varies in these assays, as they measure different apoptotic indicators. The antiapoptotic activity of this miRNA was reconfirmed by abrogating this miRNA through its sequence-specific inhibitor. 

The regulation of apoptosis by miRNAs gives an edge to HCMV besides its proteins, as they are non-immunogenic and expressed in both the lytic and latent phase of infections. Multiple studies show that the HCMV miRNAs play a regulatory role on cellular apoptosis. The HCMV miRNAs, hcmv-miR-UL148D, and miR-UL36-5p inhibit apoptosis by the translational repression of cellular genes, i.e., IEX1 and ANT3, respectively in HEK293 cells [9,11]; hcmv-miR-UL36-3p, miR-US25-2-3p, and miR-UL22A-3p downregulate the apoptosis by targeting Caspases 3 and 7 in Human Foreskin Fibroblasts [12]; hcmv-miR-US5-1 and miR-UL112-3p prevent the viral-induced apoptosis by targeting the transcription factor FOXO3a, which in turn reduces the expression of proapoptotic gene BCL2L11 in CD34+ hematopoietic progenitor cells [13]. In addition to the inhibition of apoptosis, few HCMV miRNAs reported to upregulate the apoptosis. The hcmv-miR-US25-1-5p aggravates ox-LDL induced apoptosis in endothelial cells targeting and downregulating BRCC-3 [24]; hcmv-miR-US4-1 promotes apoptosis in HCMV-infected HELF cells by targeting the Glutaminyl-tRNA Synthase (QARS) [25]; hcmv-miR-US4-5p promotes apoptosis by downregulating the transcript and protein of p21-activated kinase 2 in HEK293, HELF, and THP-1 cells [26]. Therefore, it is imperative that the regulation of apoptosis by HCMV miRNAs is dual in nature, and HCMV uses this regulation in a more intricate manner so that the outcome will benefit the virus. 

In our earlier *in silico* studies, we speculated the antiapoptotic role of hcmv-miR-UL70-3p, predicting that it can translationally repress the proapoptotic gene, MOAP1 [10]. The current *in vitro* studies also show the antiapoptotic effect of this HCMV miRNA on H_2_O_2_-induced apoptosis. So, we examined the effect of hcmv-miR-UL70-3p effect on the expression of MOAP1 mRNA during the H_2_O_2_-induced apoptosis in HEK293T cells. The results show that the ectopic expression of this miRNA significantly downregulated the MOAP1 mRNA and the apoptosis in HEK293T cells (Figure 2b). Furthermore, we verified that the miRNA binds to the 3′UTR of MOAP1, as predicted in our *in silico* studies using the dual luciferase reporter assays containing both the wild/deleted 3′UTR MOAP1 constructs. The dual luciferase reporter assays confirm the functionality of the binding site as predicted in our *in silico* studies (Figure 2 d), suggesting that MOAP1 is a functional target for the hcmv-miR-UL70-3p. 

The downregulation of apoptosis by the miRNAs through the translational repression of the MOAP1 was reported earlier by Yan and Zhao in 2012 [27] and Wu et al. in 2015 [28]. In these studies, the cellular miRNAs hsa-miR-1228 and hsa-miR-25 reported to prevent the cellular apoptosis by targeting the 3′UTR of MOAP1 mRNA and thereby repressing the MOAP1 protein [27,28]. The MOAP1 is first reported as a mitochondria enriched 39.5 kDa molecule, which rapidly upregulates in response to apoptotic stimuli, becomes associated with Bax (Bcl2 associated X), and induces the mitochondrial apoptosis [29]. Upon apoptotic stimuli, the MOAP1 associates with Bax, induces a confirmational change, and translocates Bax on the mitochondrial membrane, resulting in the release of apoptogenic factors such as Cytochrome C, initiating the mitochondrial-dependent apoptosis [30,31]. So, the translational repression of MOAP1 by the hcmv-miR-UL70-3p results in the downregulation of H_2_O_2_-induced apoptosis. 

As discussed earlier, besides the hcmv-miR-UL70-3p, two other human miRNAs also regulate apoptosis by translational repression of the MOAP1 by binding to the 3′UTR of MOAP1. So, we analyzed the binding sites for these three miRNAs in the 3′UTR region of MOAP1 and found they are different (Appendix A), and these miRNAs also do not share any sequence homology among them (Appendix A), suggesting that multiple miRNAs target the MOAP1 and regulate the apoptosis. The comparative translational repression activities between the hcmv-miR-UL70-3p and the specific siRNA for the 3’UTR of MOAP1 were studied, and it was found that although both can downregulate the mRNA and protein levels of MOAP1, the siRNA exhibits a higher rate of inhibition (Figure 3c,d). 

The regulation of apoptosis by HCMV is more complex and spatiotemporal, utilizing its proteins, non-coding RNA, and miRNAs. The majority of the studies indicate that the HCMV targets majorly the mitochondrial-dependent intrinsic pathway of apoptosis, as it may be advantageous for survival inside the cell. The HCMV protein, pUL37x1/vMIA [32], β2.7 RNA [7], and hcmv-miR-UL36-5p [11] inhibit apoptosis by disrupting the mitochondrial network. The present study adds one more HCMV miRNA-mediated regulation to the mitochondrial-dependent apoptosis.

## 4. Materials and Methods

### 4.1. Cell Culture

Human Embryonic Kidney 293T (HEK293T) cell lines were procured from American Type Cell Culture (ATCC, CRL-3216) and grown in Dulbecco’s Modified Eagle’s medium (DMEM) (Gibco) supplemented with 10% (*v*/*v*) fetal bovine serum (FBS) (Gibco, Brazil origin) and 1% antibiotic and antimycotic (Gibco). Cells were incubated in CO_2_ incubator at 37 °C with 5% CO_2_ and were passaged at 80% confluency. The cells were routinely tested for any contaminations and mycoplasma infections. The cells were divided into 4 different experimental groups (Group 1: negative control (without any treatment); Group 2: positive control (H_2_O_2_ treated); Group 3: test group, cells transfected with 25 nM of hcmv-miR-UL70-3p mimic followed by H_2_O_2_; and Group 4: cells transfected with miR-UL70-3p mimic along with its sequence-specific inhibitor followed by H_2_O_2_ treatment).

### 4.2. Apoptosis Induction and Evaluation

The apoptosis is induced in the cells using hydrogen peroxide (H_2_O_2_). Briefly, HEK293T cells were divided into four different groups as mentioned above, seeded with cell density @ 0.3 × 10^6^ for a 6-well plate and 0.01 × 10^6^ for a 96-well plate, the 3rd group was transfected with 25 nM hcmv-miR-UL70-3p, and the 4th group was transfected with an equimolar concentration (25 nM) of hcmv-miR-UL70-3p and its inhibitor. After 24 h of incubation, the 2nd, 3rd, and 4th groups were treated with 0.4 mM H_2_O_2_ and incubated for 5 h, and the 1st group is without H_2_O_2_ treatment. Then, all these cell groups were examined for apoptosis induction and inhibition through chromatin condensation, nuclear fragmentation (DAPI staining), membrane ruffling (SEM), apoptotic cells (flow cytometry), and Caspases 3/7 levels (Caspase Glo 3/7 assay). 

### 4.3. Hcmv-miR-UL70-3p Mimic, Its Inhibitor, siRNA of MOAP1, and UTR Vector Constructs of MOAP1

The hcmv-miR-UL70-3p mimic (5´GGG GAU GGG CUG GCG CGC GG 3´; Assay ID: MC11312; Cat No: 4464066) and its sequence-specific inhibitor (Assay ID: MH11312; Cat No: 4464084) were procured from Ambion, Life Technologies through Thermofisher Scientific. The siRNA of MOAP1 (sequence 5’ AGU AUA UAG ACU GUU CUA CCU UCA UGC 3’; Design ID: hs.Ri. MOAP1.13.1) designed against the 3’UTR of MOAP1 at the position of 38 to 66 nt was procured from the Integrated DNA Technologies, Inc., Coralville, IA, USA. The 3’ UTR of MOAP1 (990 bp) cloned in the pEZX-MT06 vector (Cat No: HmiT016794-MT06) and without the 3’UTR of MOAP1 (Cat No: Cmi000001-MT06) were procured from Genecopoeia Inc., Rockville, MD, USA. 

### 4.4. 4′,6-Diamidino-2-phenylindole (DAPI) Staining

To quantify the chromatin condensation and nuclear fragmentation in the different experimental groups of cells, DAPI staining was carried out according to the method described by Chazotte in 2011 [33]. Briefly, the different experimental groups of cells mentioned above were grown on the sterile cover slips fixed with 3.7% formaldehyde and then permeabilized with Triton-X 100 (0.2% in PBS). Cells were stained with DAPI in PBS (1 μg/mL) and allowed to stand for 5 min in dark condition. Finally, the images were captured using the blue filter at an excitation maxima value of 358⁄461 nm through Confocal Microscope (LSM 900- Carl Zeiss-Germany). 

### 4.5. Scanning Electron Microscopy

The membrane blebbing/ruffling was observed in the said groups of cells through scanning electron microscope. Briefly, the HEK293T cells were seeded at a density of 0.15 × 10^6^ cells/mL on to 22 × 22 mm glass coverslips (Blue Star, Mumbai, India) supplemented with DMEM and incubated to attain 40% confluency. Then, the cells were transfected with 25 nM hcmv-miR-UL70-3p and/or its inhibitor; after 24 h, they were treated with 0.4 mM H_2_O_2_ (70% confluency). After 5 h of incubation, cells were fixed in 2.5% glutaraldehyde for 4 h at 4 °C and post fixation with 1% osmium tetraoxide. Then, the cells were dehydrated with acetone with a gradual increment of concentrations (30, 50, 70, 90, 95, and 100%). Then, the cells were mounted on aluminum stubs with carbon tape and were coated with platinum using a sputter coater (JFC 1600; JEOL) at 20 mA. Then, the images were captured at 800× and 4000× magnifications using the scanning electron microscope (JOEL-JSM6490LV). 

### 4.6. Analysis of Cell Nuclear Morphology and Membrane Ruffling with ImageJ

To identify the changes in the nuclear morphology and the membrane ruffling during the induction and inhibition of H_2_O_2_-induced apoptosis, the different experiential groups of cells nuclei stained with DAPI and the cells SEM photographs obtained at 800× magnifications were assessed through ImageJ (ver 1.53e) software as described by Eidet et al., 2014 [34]. Briefly, the 16-bit images of DAPI stained nuclei were converted to 8-bit images; then, they were auto-thresholded by the “make binary” function using the default method. Then, the nuclei touching was separated with the “watershed” function and analyzed by nuclear morphology, excluding the nuclei residing on the edges of the image through the “exclude on edges” function, and small nuclei fragments were omitted from the measurements. By comparing the total number of cell nuclei obtained with or without the use of waterhead, an estimate range was computed for the association percentage between cell nuclei. 

For the membrane ruffling, the SEM images at 800× were processed similarly in the ImageJ software; then, the 3D surface plots were generated by using the option “surface plot” in the “analysis” table. Based on the surface plot, the histograms showing mean cell surface coverage area were plotted.

### 4.7. Flow Cytometry

The apoptotic cells were measured in the different experimental groups of cells by using Annexin-V conjugated with Alexa Fluor 488 (Cat No: A13201; Life Technologies) and propidium iodide (PI) (Cat No: P1304MP; Invitrogen) using a flow cytometer (BD FACS Canto). Briefly, the cells were seeded in a 6-well plate, and after 24 h of incubation, they were transfected with hcmv-miR-UL70-3p and hcmv-miR-UL70-3p along with its inhibitor, which was followed by H_2_O_2_ treatment. The group of cells without transfection of hcmv-miR-UL70-p and/or its inhibitor, with and without H_2_O_2_ treatment, serve as negative and positive controls. After 5 h of incubation, the cells were washed with cold PBS, resuspended in 100 μL of 1× binding buffer, and incubated with 5 μL of Annexin-V and 10 μL of PI in dark for 15 min at room temperature. Then, after being incubated for 15 min at room temperature, they were analyzed by flow cytometer. 

### 4.8. Caspase 3/7 Assay

The apoptotic induction and inhibition in the different groups of cells mentioned above were assessed by measuring the Caspases 3/7 levels using a Caspase-Glo 3/7 assay kit (Cat No: G8090; Promega, Madison, WI, USA) as per the manufacturer’s instructions. The raw luminescence values were detected using the luminometer (GloMax^®^ Navigator System, Promega).

### 4.9. qRT-PCR

Total RNA was isolated from the HEK293T cells by a Pure Link RNA mini kit (Cat No: 12183018A; Invitrogen, Carlsbad, CA, USA,) according to the manufacturer’s instructions. The cDNA was prepared using 1 µg of total RNA through the Proto Script^®^ II First Strand cDNA Synthesis Kit (Cat no: E6560S; New England Biolabs, Rowley, MA, USA). qRT-PCR was carried out using SYBR green chemistry by using PowerUp™ SYBR™ Green Master Mix (Cat No: A25742; Applied Biosystems, Waltham, MA, USA). The forward and reverse primers for the quantitative analysis of MOAP1 expression were 5′-CACGAGCACTAGATCACGGCTGCTGGA-3′ and 5′-CTGCCACACAGCAGCTCTGGGAGATGCC-3′, respectively [27]. The GAPDH mRNA fragment was amplified as an internal control using the forward-5′- ACATCGCTCAGACACCATG-3′ and reverse- 5′- TGTAGTTGAGGTCAATGAAGGG-3′ primers, respectively (IDT Technologies, Coralville, IA, USA). The expression of hcmv-miR-UL70-3p was detected by using the forward primer 5′-GGGGATGGGCTGGCGC-3′ and reverse 5′CTCAACTGGTGTCGTGGA-3′, respectively. The internal control is 5S rRNA, which is amplified through the primers 5′-GTCTACGGCCATACCACCCTGAAC-3′ and reverse 5′-CTCAACTGGTGTCGTGGA-3′ respectively, as mentioned in Shen et al., 2014 [21]. The qRT-PCR was performed with the following conditions: UDG activation at 50 °C for 2 min, Hot start (Activation of SYBR Green) at 95 °C for 2 min followed by denaturation at 95 °C for 15 s, annealing at 60 °C for 1 min for 40 cycles along with the melt curve analysis as per the manufacturer’s protocol. The relative mRNA expression was identified after normalizing the expression value of the test with the GAPDH in the corresponding sample by 2^-ΔΔCt^. The measurement was done in triplicate, and the results are presented at the mean ±SEM. 

### 4.10. Dual-Luciferase Reporter Assay

Luciferase vector constructs of 3’UTR of MOAP1, both wild pEZX-MT06-3’UTR-MOAP1^WT^ (Cat No: HmiT016794-MT06) and deleted type pEZX-MT06-3’UTR-MOAP1^DEL^ (Cat No: CmiT000001-MT06), were procured commercially from Genecopoeia, Rockville, MD, USA, which contains firefly luciferase as a reporter gene and renilla luciferase as the tracking gene. The HEK293T cells were co-transfected with 1 μg of vectors along with the 25 nM of hcmv-miR-UL70-3p mimic and equimolar concentrations of hcmv-miR-UL70-3p mimic along with its inhibitor using lipofectamine 3000 (Cat No: L3000008; Invitrogen). After 24 h of incubation, both firefly and renilla luciferase activities were measured by using the Dual Luciferase Reporter Assay (Cat No. E1910; Promega Corporation, Madison, WI, USA). The measurements were done in triplicate, and the maximum luciferase activity was calculated by normalizing the firefly luciferase activity to renilla luciferase activity within each sample. The results were presented as the mean ± SEM.

### 4.11. Western Blotting

MOAP1 protein downregulation by hcmv-miR-UL70-3p was assessed by performing Western blotting. The total cellular protein from the different groups of samples (HEK293T cells) was extracted with the RIPA cell lysis buffer as per the standard protocol. Protein quantification was done with the Bicinchoninic Acid (BCA) Protein Assay kit (Cat No: 786-570; G Biosciences, St. Louis, MO, USA). Equal concentrations of protein samples from the defined experimental groups were separated in 10% polyacrylamide gel and transferred onto the Immobilon-PVDF membrane at 4 °C (Millipore, Billerica, MA, USA). The PVDF membrane was incubated with the primary antibody of MOAP1 (Cat No: H00064112-M02; Novus Biologicals) and its HRP-conjugated secondary antibody (Cat No: sc-516102; Santa Cruz Biotechnologies), β-actin primary antibody (Cat No: 4970S; Cell Signaling Technology), and the HRP conjugated secondary antibody (Cat No: 7074S; Cell Signaling Technology), respectively. The blots were visualized using an enhanced chemiluminescence detection kit (Amersham, GE Healthcare, Waukesha, WI, USA), and the developed blots were observed by Chemiscope (Clinx, Shanghai, China). The blots were repeated in three independent experiments, and the density of each protein band was quantified by using ImageJ software (ver 1.53e).

### 4.12. Statistical Analysis

Statistical analysis was performed using Graph Pad Prism 7.0 Software Inc. Data from three independent experiments were presented as mean ± SEM used for statistical analysis. Statistical significance was determined by using ANOVA. A *p* value of <0.05 was considered statistically significant. 

## 5. Conclusions

The regulation of apoptosis by HCMV through its proteins is a well-established phenomenon. Growing evidence shows that HCMV miRNAs took part in regulating this process and exert their effect synergistically or independently to the HCMV proteins. The current study reveals one such HCMV miRNA exerting antiapoptotic effects. The ectopic expression of hcmv-miR-UL70-3p downregulates the H_2_O_2_-induced apoptosis by translational repression of the protein modulator of apoptosis-1 (MOAP1). MOAP1 is a critical mitochondrial effector of Bax, so, targeting this protein, HCMV regulates the mitochondrial-dependent intrinsic pathway of apoptosis. 

## Figures and Tables

**Figure 1 ijms-23-00018-f001:**
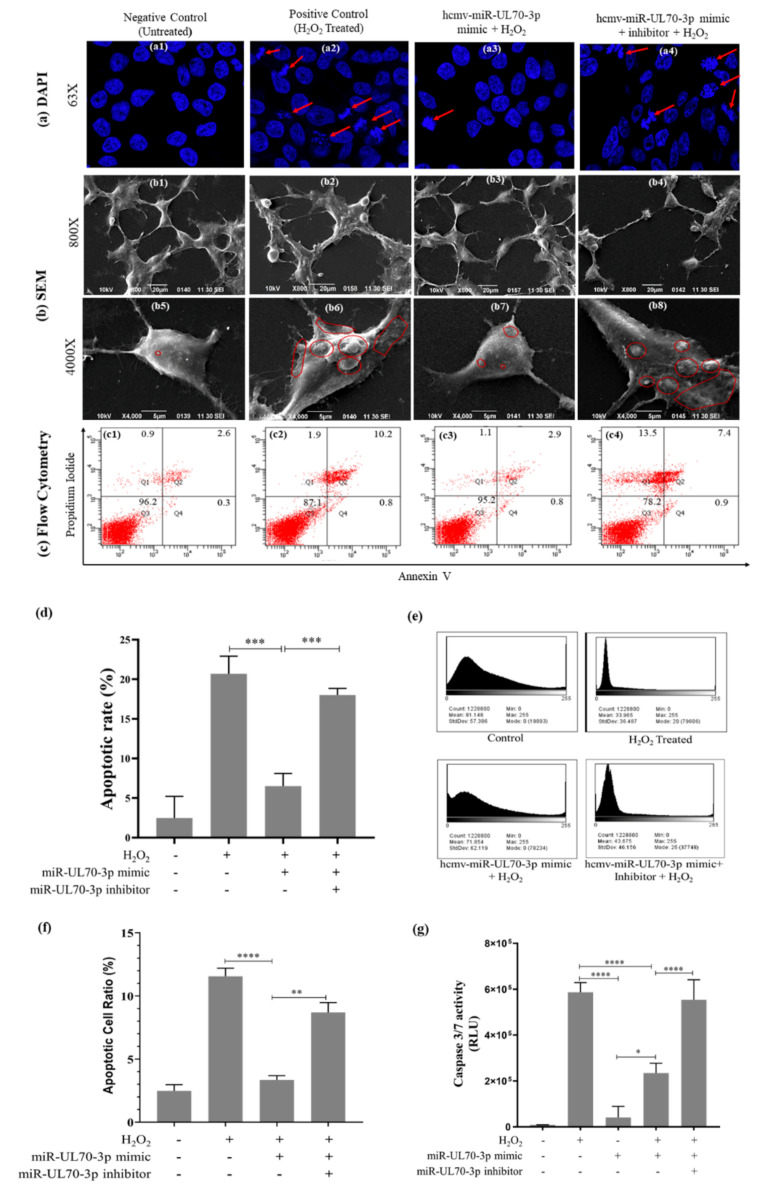
Effect of hcmv-miR-UL70-3p on H_2_O_2_-induced apoptosis. HEK293T cells were divided into four different groups to evaluate the effect of hcmv-miR-UL70-3p on H_2_O_2-_induced apoptosis, 1st group as negative control (untreated); 2nd group as positive control (treated with H_2_O_2_ only); 3rd group is transfected with hcmv-miR-UL70-3p mimic followed by H_2_O_2_ treatment, and the 4th group is transfected with hcmv-miR-UL70-3p mimic along with its inhibitor followed by H_2_O_2_ treatment. (**a**) Representative images showing the nuclear morphology of different groups of cells (63×) after DAPI staining. The apoptotic nuclei were indicated through red arrows in (**a1**–**a4**). The apoptotic rate was calculated using the equation $Apoptotic\;Rate\;\left(\%\right)=\frac{\mathrm{Number}\;\mathrm{of}\;\mathrm{apoptotic}\;\mathrm{Nuclei}}{\mathrm{Total}\;\mathrm{number}\;\mathrm{of}\;\mathrm{Nuclei}}\times 100$ and is shown in (**d**) (±SEM; ***, *p*<0.001). Data from three independent experiments were used for statistical analysis, (±SEM; ****, *p* < 0.0001). (**b**) SEM images of different cell groups taken at 800× and 4000×, and the membrane blebbing/rufflings are marked in red lines (**b1**–**b8**). The 800× images (**b1**–**b4**) were used in generating the histograms showing the mean cell surface coverage area through ImageJ Software (**e**). The representative dot plots showing live and apoptotic cells determined by flow cytometry are arranged in (**c**). (**f**) shows the apoptotic cell ratio for flow cytometric analysis, data from three independent experiments were used for statistical analysis (±SEM; **, *p* < 0.01; ****, *p* < 0.0001). (**g**) Caspases 3/7 activity measurements were plotted as RLU, and the data from three independent experiments were used for statistical analysis (±SEM; *, *p* < 0.05; ****, *p* < 0.0001).

**Figure 2 ijms-23-00018-f002:**
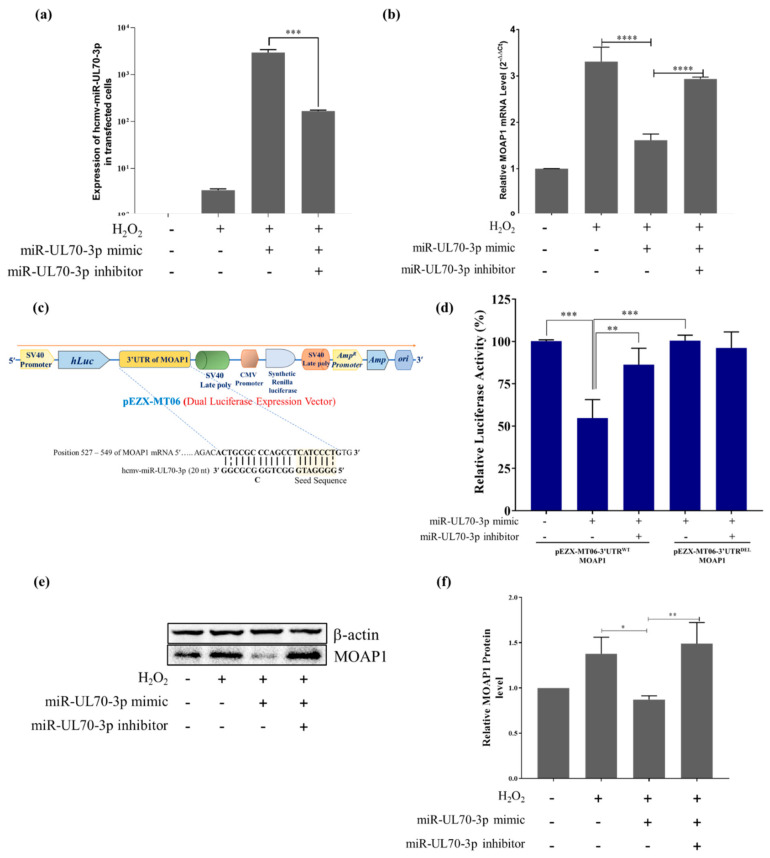
(**a**) Hcmv-miR-UL70-3p expression in the HEK293T cells post transfection with the hcmv-miR-UL70-3p mimic: The hcmv-miR-UL70-3p expression levels were measured in the cells post transfected with hcmv-miR-UL70-3p mimic. The quantity of hcmv-miR-UL70-3p levels was normalized to that of 5s rRNA in the corresponding samples. Data from three independent experiments were recorded and presented (± SEM; ***, *p* < 0.001). (**b**) Downregulation of MOAP1 mRNA by the ectopic expression of hcmv-miR-UL70-3p: The expression of MOAP1 mRNA in different experimental groups was determined by using qRT-PCR. The mRNA quantity of MOAP1 was normalized to that of GAPDH, and each sample was analyzed in triplicate. Results are expressed as the fold change (2^−^^ΔΔCt^) in treated groups relative to the control group (±SEM; ****, *p* < 0.0001). (**c**) Cartoon depicting the components of the pEZX-MT06 dual luciferase reporter vector having the 3′UTR of MOAP1, and the enlarged portion showing the binding position of hcmv-miR-UL70-3p in the 3′UTR of MOAP1. (**d**) Hcmv-miR-UL70-3p targeting the 3′UTR of MOAP1: Dual luciferase reporter assays were performed to test the interaction of hcmv-miR-UL70-3p to the 3′UTR of MOAP1 using pEZX-MT06- 3′UTR^WT/DET^-MOAP1 vector constructs. Data from three independent experiments were used for statistical analysis (±SEM; **, *p* < 0.01; ***, *p* < 0.001). (**e**) Hcmv-miR-UL70-3p downregulates the MOAP1 protein: The MOAP1 expression in the cells treated with or without hcmv-miR-UL70-3p, hcmv-miR-UL70-3p, and its inhibitor was detected by Western blotting. The bands of the blot have been cropped with no further manipulation. (**f**) The relative MOAP1 protein quantification was performed with ImageJ software, after normalizing with β-actin. Experiments were performed in triplicates (Appendix A), and the data from three different experiments were used for statistical analysis (±SEM; *, *p* < 0.05; **, *p* < 0.01).

**Figure 3 ijms-23-00018-f003:**
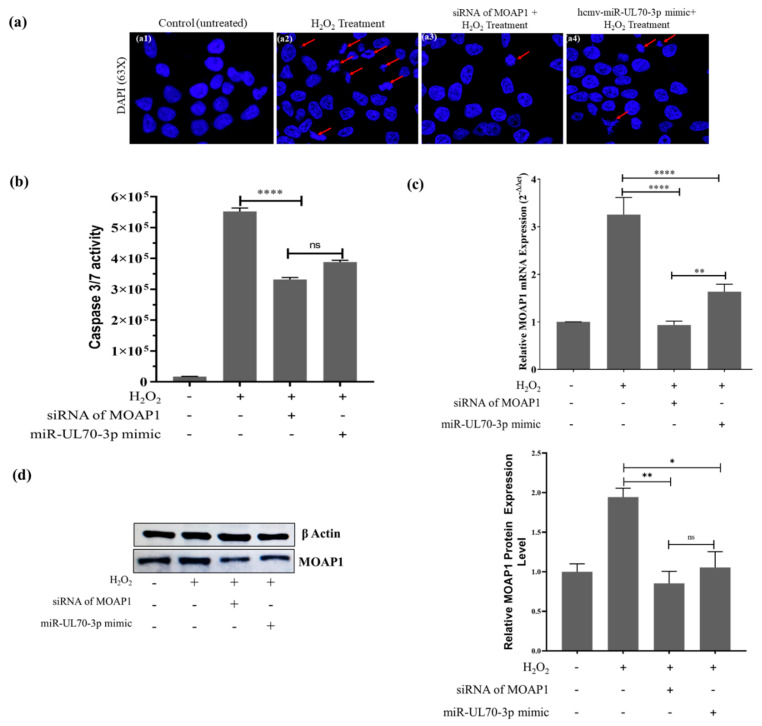
Comparison of downregulation of apoptosis/MOAP1 by siRNA of MOAP1 and hcmv-miR-UL70-3p: (**a**) Through DAPI imaging, images (63×) showing the nuclear morphology. (**b**) Measurement of Caspase 3/7 activity: The Caspase 3/7 activities were measured through Caspase Glo 3/7 assay in four different groups of cells, as mentioned above, these were recorded as raw luminescence units (RLU) (±SEM; *****p* < 0.0001). (**c**) MOAP1 mRNA downregulation: The relative expression levels of MOAP1 mRNA after the transfection of either hcmv-miR-UL70-3p and siRNA of MOAP1 were measured through qRT-PCR. Results were expressed as the fold change (2^−ΔΔCt^) (± SEM; **, *p* < 0.01; ****, *p* < 0.0001). (**d**) MOAP1 protein downregulation: The MOAP1 protein downregulation after transfection with either siRNA of MOAP1 and hcmv-miR-UL70-3p were analyzed through Western blot. The relative MOAP1 protein quantification was performed through ImageJ software after normalizing with β-actin. Experiments were performed in triplicates (Appendix A), and the data from three different experiments were used for statistical analysis (±SEM; *, *p* < 0.05; **, *p* < 0.01, ns = non-significant).

## Data Availability

Supporting data are included as electronic Appendix A.

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
