# Peer review of "Human Cytomegalovirus miR-UL70-3p Downregulates the H2O2-Induced Apoptosis by Targeting the Modulator of Apoptosis-1 (MOAP1)"

_ijms, 2021, doi:10.3390/ijms23010018_

Round 1
Reviewer 1 Report
The subject matter discussed in the manuscript is interesting, despite the fact that it is a subject from a narrow, specialized scientific area. The authors presented their great knowledge and scientific skills. I do not see any significant shortcomings. My only suggestion is to improve the quality of Figures, as some descriptions are difficult to read. Font too small.
I recommend the manuscript for publication.
Reviewer 2 Report
- The authors mentioned in section of 4.8 that statistical analysis was performed, but it was not presented in the article. The author should add the statistical data analysis to strengthen the correctness of the author's opinion.
- Most of the references were not new enough. The authors should update the references and introduce more latest references in the manuscript.
In my opinion, this article can be published in this journal if the authors offer enough new data and update the references. Also, the authors should recheck and correct the errors of typos and modify the structure of the article before sending out this manuscript. By all the aspects enumerated above, the author should make a minor revision of the paper before its publication.
Reviewer 3 Report
The manuscript written by Pandeya A and co-authors presents the molecular investigation of human cytomegalovirus miR-UL70-3p effect on regulation of apoptosis by targeting the modulator of apoptosis (MOAP1). Current manuscript is an evaluation of previously published by authors study of in silico findings related to the antiapoptotic nature of hcmv-miR-UL70-3p in HEK293 cells by H2O2 inducing apoptosis. Authors report that hcmv-miR-UL70-3p targets the proapoptotic gene MOAP1 through interaction with its 3'UTRregion of mRNA. They use a variety of different assays to show the ectopic expression of hcmv-miR-UL70-3p mimic significantly downregulates the H2O2 induced apoptosis by translational repression of MOAP1. Manuscript is well written, good scientific design and results support conclusions. But, I have some specific comments:
- For figure 1 authors use 4 different groups but missed one group of cells transfected with hcmv-miR-UL70-3p with no H2O2 treatment to induce apoptosis. Transfection of miRNA's could be also trigger for apoptosis. At least for one experiment authors can make this control.
- Quality of microscopic images in Figure 1 A do not allow to identify reliably apoptotic nuclei from non apoptotic nuclei. Chromatin condensation in apoptotic cells is a very important marker for development of apoptosis and those assays are very well established in literature. In materials and methods authors didn't describe what kind of cells they use: live or fixed. I suggest to repeat this experiment on fixed cells with better imaging that allow to calculate apoptotic nuclei accurate.
Round 2
Reviewer 3 Report
Quality of confocal images showing chromatin condensations are improved, results are convincing. Methods are described in details that make conclusions supported by data.